# Ignored Opinions: Villager-Satisfaction-Based Evaluation Method of Tourism Village Development—A Case Study of Two Villages in China

**Naifei Liu** * 📧 **, Kaijian Yue and Xiaoyue Zhang**

School of Architecture, Tsinghua University, Beijing 100084, China; yuejj21@mails.tsinghua.edu.cn (K.Y.);
xiaoyue-22@mails.tsinghua.edu.cn (X.Z.)
* Correspondence: lnf20@mails.tsinghua.edu.cn

**Abstract:** The neglect of endogenous strength is one of the reasons for the lack of sustainability in mountainous rural development and tourism development in China at present. How to incorporate the opinions of villagers in the tourism development process led by the government and other external entities is the main focus of this article. Based on the fieldwork of two typical mountainous villages and a previous rural development evaluation method, this article proposes the villager-satisfaction-based evaluation method for tourism village development, covering rural settlement construction, village esthetics, and economic and social development. "Villager satisfaction" is a crucial indicator obtained by objectifying the subjective opinions of villagers. Finally, the evaluation method was applied in the form of a questionnaire in two villages. The experimental results are correlated with the tourism development patterns of the two villages, verifying the feasibility and effectiveness of the evaluation method. It is expected that this evaluation method will become an effective communication medium between non-professional villagers and the professional tourism development process, thereby promoting the sustainable development of rural areas in the future.

**Keywords:** villager satisfaction; mountainous village; tourism village development; evaluation method; sustainable development; Chinese rural area; traditional village

## 1. Introduction

### 1.1. Background

China has numerous mountainous regions, and many villages are scattered throughout hilly and mountainous areas. The geographical terrain hinders transportation connections and cultural diffusion, resulting in remote mountainous areas being less influenced by modernization and urbanization. This, in turn, has preserved the traditional customs, settlement patterns, and architectural styles of these villages. In recent years, there has been a reflection on modernization and a revival of traditional culture, leading to an increased appreciation of the value of traditional villages. As a result, traditional villages have become hotspots for tourism development.

However, for remote mountainous areas, tourism development faces multiple challenges. Firstly, inconvenient transportation in mountainous regions limits tourist visits. Secondly, due to the generally low economic levels in mountain villages, the existing resources and facilities are often insufficient to meet the needs of external tourists. Additionally, mountainous villages tend to be small in scale, with scattered sight spots, which makes management and promotion difficult and hinders the formation of clustered and large-scale tourism routes. These factors result in high initial investments and slow returns in mountainous tourism development, and a sustained investment is required to truly promote local economic development.

In contemporary China, most tourism village development projects are primarily led by the government and other external funds. The government-led governance model and

profit-driven mentality of the capital have led to a lack of sustainability in tourism village development. The ultimate cause of this phenomenon lies in the neglect of endogenous strength within rural areas. In other words, the villagers, who are the main actors in rural areas, are excluded from development decision making, making it difficult to harness some localized resources. Therefore, encouraging endogenous strength at the local level and achieving a transition from external intervention to internal spontaneous development are key to achieving sustainable development for mountainous tourism villages.

*1.2. Research Question*

Villagers are the primary source of local endogenous strength. The top-down development model in China inherently lacks attention to the opinions of grassroots villagers. Even when villagers are provided with the opportunity to express their views, most are constrained by their educational levels, resulting in fragmented and vague opinions. Relevant authorities find it challenging to incorporate these views, let alone influence the development process. Consequently, villagers perceive their opinions as unimportant, leading to a decrease in their attempts to express their views. Developers and the government, in turn, assume that villagers are disinterested in the village's development and are unlikely to provide constructive input. This detrimental cycle results in the gradual erosion of the village's intrinsic developmental drive. This issue becomes even more pronounced in the context of traditional tourism village development.

Therefore, the research question of this study is how to establish an effective means of communication between villagers and professional tourism developers, specifically by developing a village-centric evaluation method for tourism village development.

## 2. Literature Review

### 2.1. Tourism as a Catalyst for Sustainable Development in Traditional Villages

Since 2012, China's Ministry of Housing and Urban–Rural Development, Ministry of Culture, State Administration of Cultural Heritage, and Ministry of Finance have jointly initiated investigations and established a protection list for China's traditional villages. Important criteria for determining whether a village is traditional include the integrity and antiquity of existing traditional architectural styles and layouts, as well as the preservation of traditional characteristics in the village's location and structure, which may also include the active inheritance of intangible cultural heritage.

To those traditional villages in China, tourism development has become a pivotal method for augmenting residents' income and alleviating poverty. Cultural heritage, as a critical traditional resource, when examined and utilized across various dimensions, can render cultural heritage-oriented tourism development more sustainable and positively impact overall village development [1,2].

In recent years, the Chinese government has demonstrated a consistent dedication to rural areas, and tourism-based poverty alleviation has emerged as a significant measure to combat poverty in China [3]. Numerous Chinese scholars have engaged in discussions regarding various aspects of tourism development in specific traditional villages in China, encompassing the relationship between rural revitalization and rural tourism [4], public policies for traditional village tourism development, comprehensive development frameworks [5,6], the conservation and planning of traditional villages [7,8], and transformations in village spaces and residents' living environments [9,10].

Analogous to many rural areas in third-world countries, and echoing the situation in numerous traditional villages in China, scholars have analyzed the undue challenges and obstacles these nations and regions face in rural tourism development. These challenges encompass institutional and policy irrationalities in rural tourism development, unfavorable operational management, deficiencies in professional knowledge and expertise, insufficient tourism development budgets, and the residents' limited understanding of tourism development. Scholars have proposed constructive recommendations to over-

come these challenges, ultimately driving tourism development and promoting sustainable development in rural areas [11–15].

The European Union (EU) has also shown significant concern for the preservation of impoverished rural regions, known as "lagging rural regions (LRRs)", and has extended policy support to bolster the development of these villages [16]. Some scholars have noted that EU LRRs generally possess rich socio-cultural resources, which can be effectively integrated into tourism development, thus invigorating the region's progress [17].

However, scholars have also emphasized that unregulated tourism development can lead to cultural degradation, ultimately undermining the potential for sustaining the tourism industry [18]. Simultaneously, tourism development may negatively impact local social cohesion [19]. Therefore, scholars underscore that, in traditional rural development, the endogenous strength of the local community plays a pivotal role in achieving regional sustainable development [20].

### 2.2. Rural Community and Tourism Development

Tourism development can significantly contribute to the economic growth of rural areas. Economic-oriented tourism development has been positively correlated with residents' satisfaction. Nevertheless, some scholars have raised concerns, suggesting that a sole focus on economic capital may lead to unsustainable development and foster antagonistic sentiments among community residents. It is imperative to emphasize the utilization of social capital, fully exploring and harnessing the inherent potential within the community [21,22]. The concept of community-based tourism (CBT) underscores the importance of involving and empowering community residents in the development process [23,24]. Numerous case studies from various countries underscore the pivotal role of community support in the successful execution of rural tourism development [25,26]. Community participation not only fosters tourism development but also enables a people-centered approach to diverse, sustainable development [27]. This has been well demonstrated in Japan and Taiwan, where the fundamental principle of "villager-led" movements has been introduced to transform traditional villages into modern communities, effectively highlighting the substantial role played by the community's endogenous strengths in advancing the sustainable development of these traditional villages [28,29].

To facilitate resident engagement in tourism development, a fundamental understanding of tourism's impact on the local residents is necessary. Discussions surrounding the effects of tourism on community residents date back to the 1960s. With evolving research, the assessment of its impact has transitioned from unidimensional evaluations to more multifaceted and individual-focused analyses [30,31]. For instance, scholars have delved into residents' attitudes towards tourism development and external tourists, providing insights from behavioral and emotional perspectives regarding the relationship between local residents and tourism development [31,32]. Additionally, some researchers have compiled comprehensive summaries of the challenges and restrictions faced by communities in developing countries who engage in the tourism industry, proposing constructive solutions and recommendations [33–35]. Furthermore, certain scholars have analyzed the involvement of community strengths, exploring how residents' participation in tourism development could be enhanced by comparing policies [23] and tourism management models [12]. Finally, a particular group of scholars has taken a rights-based approach to scrutinize the impact of tourism development on the local villagers through in-depth individual interviews, demonstrating that the benefits brought about by tourism development are not evenly distributed among all participants. This highlights the current issues within the community participation model of tourism development [36].

Decision-making processes and benefit allocation are the cornerstones of community-based development [37]. However, in the context of China, most development projects are primarily driven by government initiatives and external capital. Professional companies typically manage tourism projects. The imbalance between urban and rural development often results in villagers seeking employment opportunities in cities rather than remaining

within their villages. Consequently, community residents exhibit lower levels of participation in the decision-making processes and benefit allocation associated with practical tourism development.

*2.3. Village Satisfaction and Tourism Village Development Evaluation Method*

Numerous studies have attempted to understand residents' opinions on tourism development from various angles, but the assessment of residents' satisfaction with tourism development is often approached from the perspective of the overall tourism industry development [38–40]. However, most research stops at understanding "public opinion" or merely provides assessments and improvement suggestions for selected subjects [41–43]. There is a lack of systematic and widely applicable methods to incorporate these opinions into a feedback system and make specific adjustments to development plans.

In China, traditional villages have been long recognized by architectural, archeological, and artistical specialists as cultural relics [7]. Therefore, the physical environment construction is the main component of rural tourism development. Especially in recent years, with the political policy of tourism poverty alleviation implemented in China, tourism development resembles village-built environment development [6]. Thus, the existing literature on traditional Chinese rural tourism development has largely focused on the field of architecture. For the evaluation of the built environment, the field of architecture utilizes well-established theoretical frameworks, like architectural programming and post-occupancy evaluation theory [44,45]. This theory offers a rational and scientific method for assessing buildings, optimizing decision-making processes in construction. Moreover, it has been further applied to the evaluation of rural development. For example, Dang proposed a framework for rural architectural programming and post-occupancy evaluation, delving into the social principles underpinning rural architectural programming and discussing specific operational methods [46,47]. After many years of development, the evaluation method of rural construction in China has gradually become more in-depth and refined [48]. Feedback and influence on rural development via evaluation indicators play a critical role in minimizing decision-making errors during the project execution. However, the existing evaluation indicators for rural development often overlook the importance of considering residents' opinions, neglecting the significant role of community participation in sustainable rural development. Therefore, there is an urgent need to establish an evaluation method that reflects the residents' opinions to promote their involvement in the development of tourism-oriented rural areas.

## 3. Materials and Methods

*3.1. Determination of the Experimental Subjects*

3.1.1. The Jiufeng Mountain Area in Northeastern Fujian, China

This article selected the Jiufeng Mountain area (Figures 1–3) as the research area, primarily for the following two reasons:

- The region is a typical remote mountainous area with rolling terrain, dense forests, and an extensive network of waterways. Due to the transportation difficulties, it has experienced the impact of rapid urbanization in China to a lesser degree and in a more delayed manner. Like most mountainous areas, the traditional settlement patterns and architecture were preserved and have become valuable tourism resources in the present.
- With the development of rural tourism, villages in this region have become popular tourist destinations for surrounding cities, thanks to their beautiful natural landscapes and well-preserved traditional features. Pingnan County, as an epitome of tourism village development in Fujian Province, was successfully transformed from an impoverished county into a trending tourist destination. By 2022, Pingnan County, as a small county with a permanent population of only 139,000 inhabitants, built 16 "Gold Medal Tourist Villages" and welcomed 4.9 million tourists and earned CNY 4.05 billion (CNY 1 is approximately equal to USD 0.14 (based on exchange rates in September 2023)) of tourism income in that year.

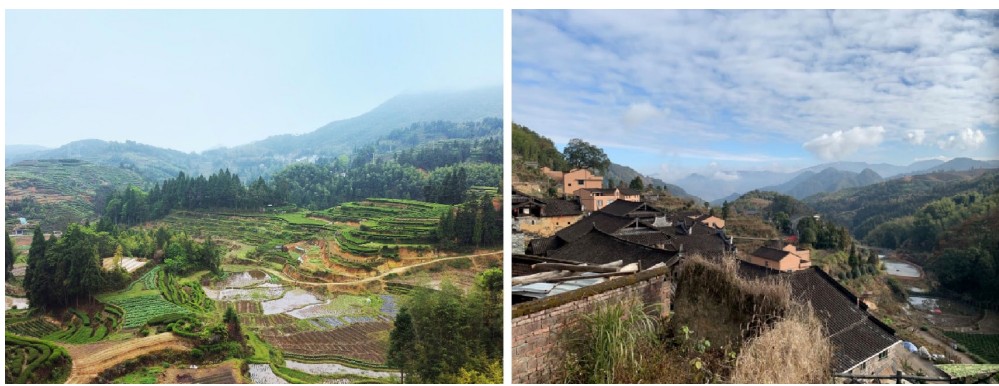

**Figure 1.** The Jiufeng Mountain area (image source: taken by the author).

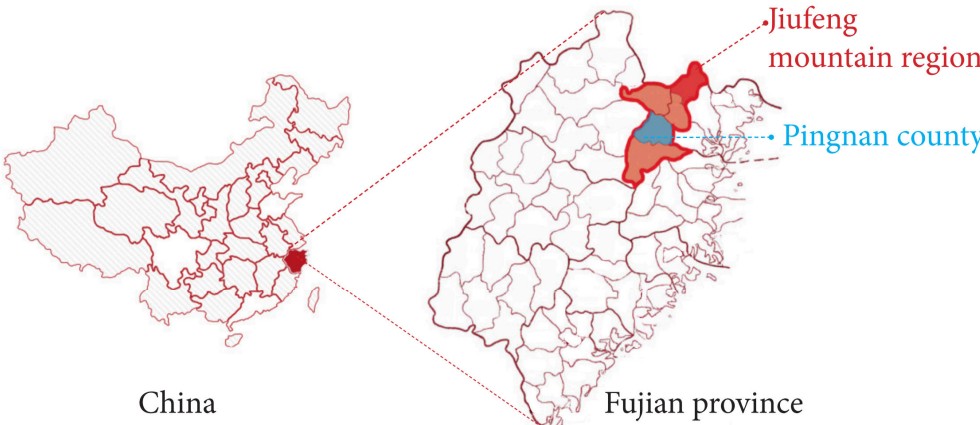

**Figure 2.** The geographical area of Jiufeng Mountain and the location of Pingnan County (image source: Drawn by the author).

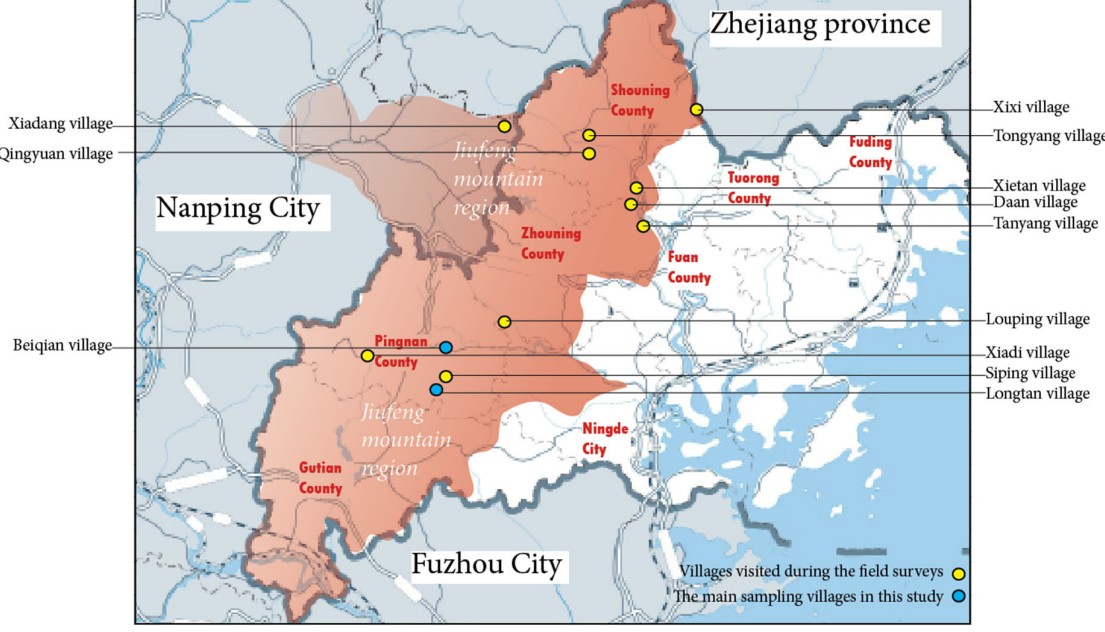

**Figure 3.** The locations of the villages visited during the field surveys in the Jiufeng Mountain area (Image source: created by the author).

### 3.1.2. Beiqian Village and Longtan Village

This article selected Beiqian Village and Longtan Village in Pingnan County as the primary observation and sampling areas (Table 1 and Figure 4), primarily based on the following two reasons:

- Both villages are the most well-known tourist destinations in the surrounding area, and they have similar transportation conditions, permanent populations, and levels of tourism development. Although there is a significant difference in the overall area, the core tourism areas are roughly equivalent. Additionally, the original villagers still live in the villages and maintain traditional lifestyles in spite of tourism developing.
- The two villages have distinct differences in their tourism development paths. By selecting these two villages as the research subjects, it is possible to first conduct a correlational analysis between the evaluation results and the actual situations in each village. Secondly, a comparative analysis of the evaluation results from the two villages can be performed to effectively test the feasibility of the evaluation method.

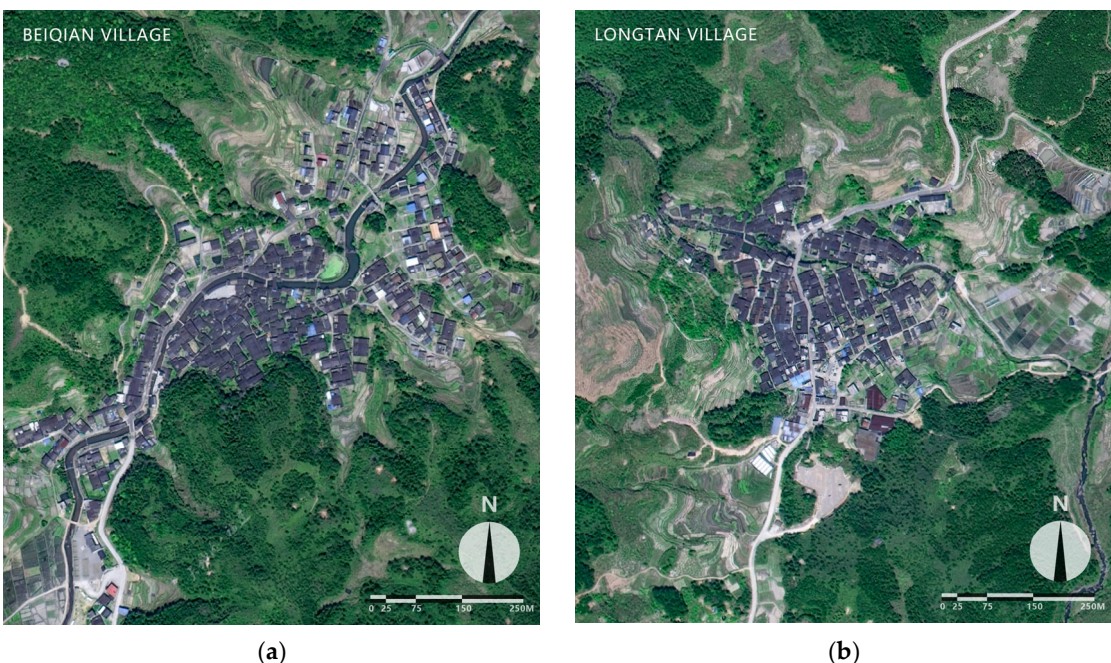

(**a**)                                                                                                      (**b**)

**Figure 4.** (**a**) Site plan of Beiqian village; (**b**) site plan of Longtan village (Image source: satellite images obtained from www.tianditu.gov.cn (accessed on 16 September 2023); redrawn by the author).

**Table 1.** Basic information regarding Beiqian Village and Longtan Village.

| Information | Beiqian Village | Longtan Village |
|---|---|---|
| Location | Daixi Town, Pingnan County, Fujian Province | Xiling Town, Pingnan County, Fujian Province |
| Distance from the city [1] | 42 km, 60-min drive | 36 km, 50-min drive |
| Registered population | 2268 | 1174 |
| Permanent population | 1280 | 814 |
| Total area | 28 km$^2$ | 5.6 km$^2$ |
| Built-up area | 67 hectares | 12 hectares |
| Tourism area | 15 hectares | 11 hectares |
| Honorary title | The third batch of Traditional Chinese Villages (2014) | The sixth batch of Traditional Chinese Villages (2023) |
| Formation time | Early 14th century | Early 15th century |
| Start of tourism development | 2016 | 2017 |

**Table 1.** *Cont.*

| Information | Beiqian Village | Longtan Village |
|---|---|---|
| Main industry | Yellow wine and cultural tourism | Cultural and creative industries |
| Development model | Government financial support and university cooperation | Government as the lead, with artist guidance and villagers' participation |
| Innovative policy | / | "15-year-lease subscription of old house", "labor and material law", and "Longtan green card" |
| Tourism in 2022 | Tourists: over 40,000 Income: over CNY 0.6 million | tourists: over 300,000 Income: over CNY 13 million |
| PCDI [2] in 2022 | CNY 23,000 | CNY 24,800 |
| Other non-agricultural industry | Yellow wine: over 1000 tons per year Income: over CNY 30 million | / |

[1] Distance from Pingnan county (the nearest city). [2] PCDI: per capita disposable income.

Beiqian Village was designated as a Chinese traditional village in 2014 by the Ministry of Housing and Urban–Rural Development of China and began its tourism development slightly earlier than Longtan village. Since 2016, it has focused on the "Yellow Wine + Cultural Tourism" industry, investing CNY 12 million in infrastructure and supporting facilities with government support. By regularly hosting large-scale events, like the Yellow Wine Culture Festival, and collaborating with universities to develop cultural and creative products, it has continuously increased its tourist attraction. In 2022, it received nearly 40,000 visitors, generating a comprehensive tourism income exceeding CNY 6 million. Meanwhile, the annual production of yellow wine is over 1000 tons, with an annual output value of over CNY 30 million. However, the homestay industry is limited, with only eight homestays and 105 beds.

The reconstruction of the built environment in Beiqian Village mainly focuses on the improvement of sanitation and landscape, and the reconstruction of the existing traditional buildings in the village is minor, striving to minimize interferences with the traditional layout and style. Figure 5 compares the situation before and after the reconstruction of the local outdoor space in Beiqian Village.

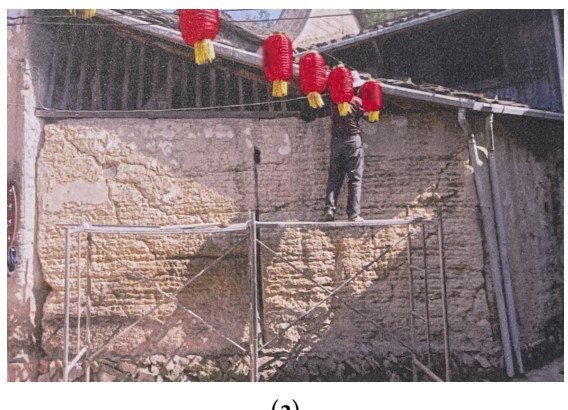 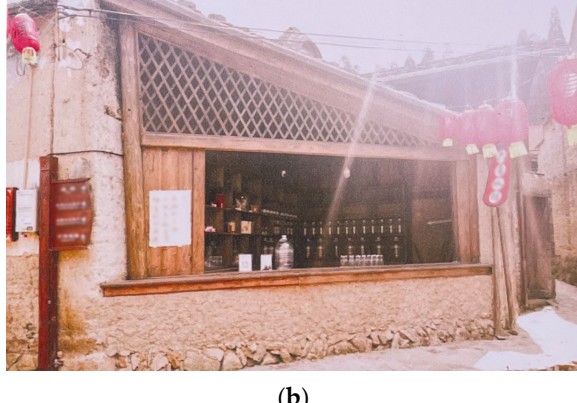

(**a**) (**b**)

**Figure 5.** One of traditional building in Beiqian Village. (**a**) The building in the village before the renovation; (**b**) the building in the village after the renovation (image source: Wu Shandi's photograph (The use of these photographs was authorized by Wu Shandi, a member of the village leaders of Beiqian Village, 2023)).

Longtan Village's tourism development began in 2017, focusing on the cultural and creative industry. It introduced contemporary artistic creativity into the ancient village by means of "government guidance + artist leadership + villagers' participation", thereby enhancing the quality of the living environment in the ancient village. Simultaneously, the

innovation of the old house renting policy allows the "new villagers" ("new villagers" is a term commonly used in recent years to refer to a group of people who have relocated from cities to rural areas to settle down or start businesses. They are distinguished from the indigenous residents who have lived in the village for generations.) to lease entire old traditional houses for a very low rent. And, in return, the "new villagers" should invest in the renovations of these houses, and these renovations should be focused on maintaining the traditional architectural esthetics. This approach allowed the transformation of these houses into various formats, such as homestays, exhibition halls, and bars. In 2022, Longtan Village received over 300,000 visitors, generating a tourism income of over CNY 13 million and attracting more than 400 "new villagers" and returning job-seekers. Over 20 homestays and more than 30 art-sharing spaces were established. It is commendable that the ecological environment of Longtan Village has been continuously improved rather than destroyed while accepting increasing tourist numbers, and traditional buildings have been better protected due to the increase in economic income.

Many renovation activities of the traditional buildings were conducted during the tourism development stage, and some new landscape buildings were built, but most of them follow the traditional style. Figure 6 compares the situation of the river landscape in Longtan Village before and after reconstruction.

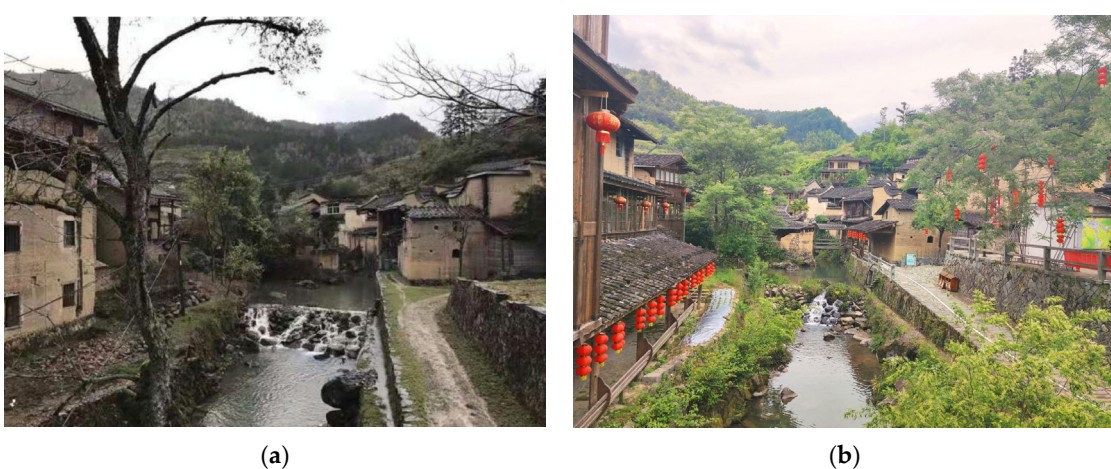

(**a**)  (**b**)

**Figure 6.** River landscape in Longtan Village. (**a**) Before tourism development (image source: Chen Xiaozhen's photograph (The use of this photograph was authorized by Chen Xiaozhen, a member of the village leaders of Longtan Village, 2023)). (**b**) After tourism development (image source: Xia Xingyong's Master thesis [49]).

In comparison, the primary stakeholders in the tourism development of Beiqian Village are the government and the village collective, and their improvement and renovation projects are more holistic. On the other hand, the tourism development in Longtan Village is more decentralized, with multiple stakeholders involved, resulting in a unique and successful tourism village development case.

### 3.2. Existing Evaluation System of Villager Satisfaction for Rural Human Settlements

Based on Shi's review of the rural human settlement evaluation research conducted over the past two decades and the evaluation systems from 60 main studies [12], the common evaluation indicators were obtained (Table 2), including a total of 8 primary indicators and 26 secondary indicators. Among the secondary indicators, 16 indicators were related to physical factors, including living conditions (6), infrastructure (6), and ecological environment (4), while 10 indicators were related to social factors, encompassing public services (3), quality of life (2), social circumstance (2), social culture (2), and economic development (1).

**Table 2.** Common evaluation indicators in rural human settlement evaluation research.

| Primary Indicators | Secondary Indicators |
| --- | --- |
| Living conditions (6) | Per capita house area<br>Day light and house orientation<br>Quality of construction<br>Architecture design<br>Density of buildings<br>Courtyard landscape |
| Infrastructure (6) | Road and transportation<br>Water supply<br>Traffic convenience<br>Electricity supply<br>Waste disposal<br>Quantity and quality of public toilets |
| Ecological environment (4) | Air quality<br>Plants and gardening<br>Sewage disposal<br>Quality of drinking water |
| Public services (3) | Shopping convenience<br>Quantity and quality of schools<br>Recreational facilities |
| Quality of life (2) | Cultural and sports facilities<br>Children's access to school |
| Social circumstance (2) | Social security<br>Health care |
| Social culture (2) | Neighborhood relationship<br>Quality of democracy |
| Economic development (1) | Per capita disposable income |

Nevertheless, according to the existing rural architectural programming and post-occupancy evaluation theory, "society", as one of the six primary elements of its programming method system, contains four secondary elements, "public participation, public opinion and social satisfaction", "social demonstration", "community relations and social progress", and "social equity" [18]. If the assessment of the pertinent factors in the evaluation system is not detailed enough, the limited information cannot be effectively utilized for the future programming or to establish a closed loop from architectural programming to post-occupancy evaluation". Therefore, it is necessary to further refine the indicators associated with social factors and advance the sophistication of the evaluation system from a theoretical perspective.

### 3.3. Exploration of Evaluation Indicators Based on Field Surveys

In order to better understand the situation of tourism village development and villagers' feedback, field surveys were conducted several times from 2022 to 2023 in the Jiufeng Mountain area of Fujian Province. A total of 12 tourist traditional villages were visited (Figures 3 and 7), and semi-structured interviews were conducted with 19 representatives, including indigenous villagers, new villagers, village officials ("village officials", sometimes called "village leaders" or "village cadres", perform the functions of exercising public power, managing public affairs and providing public services in the village. In China, they are elected by villagers and usually have a high reputation among villagers.), local experts, and government personnel, to understand their views on the development of rural tourism (Table 3). The survey found that, in addition to the construction of the physical environment, non-physical factors, such as the improvement of income, interpersonal harmony, government and village committee diligence, and the congeniality of the

business environment, play pivotal roles in determining villager satisfaction levels with tourism village development.

**Table 3.** Identities of the interviewees and the main subjects of the interviews.

| Type | Interviewee | Identity | Main Subject of Interview |
|---|---|---|---|
| Government officials and local experts | Wan | Fujian Province government | Policy logic of rural tourism development |
| | Song | Ningde Tourism Management Association | Application of and support for Traditional Chinese Village |
| | Fa | Pingnan first middle school | Development of rural tourism in Pingnan |
| | Li | Shouning housing and urban–rural development bureau | Rural tourism and inheritance of traditional techniques in Shouning |
| | Ye | Qingyuan Town government | Policy of rural settlement environment improvement |
| | Gong | Qingyuan Town government | Planning of village and rural tourism |
| Village officials | Wu | Beiqian Village | History and tourism development of each village |
| | Chen | Longtan Village | |
| | Gong | Xiadi Village | |
| Indigenous villagers | A wine seller | Beiqian Village | Experience in Beiqian |
| | A middle-aged Worker | | |
| | A young entrepreneur | Longtan Village | Personal experience of running a business |
| | A tea maker | Xiadi Village | Motivation of returning to village |
| | An antique dealer | Tongyang Village | founding experience of Tongyang Cultural Park |
| New villagers | A homestay owner from Ningde | Longtan Village | Motivation of coming to Longtan and running of a homestay business |
| | A homestay owner from Fuzhou | | |
| | A volunteer | Xiadi Village | Motivation of coming to Xiadi and the circumstances of foreign volunteers |
| | A young porcelain maker | Xiadang Village | Running of traditional craft workshops |
| | A young entrepreneur | Siping Village | Work of rural revitalization workstation |

Based on the field surveys and literature review, the evaluation indicator system was formulated, as illustrated in Table 4, including 5 primary indicators and 19 secondary indicators. Among the secondary indicators, there are 1 measure for overall satisfaction with tourism development, 6 indicators for satisfaction with physical environment construction, 4 indicators for satisfaction with rural scene, 3 indicators for satisfaction with economic and social development, and 5 indicators for satisfaction with social relations.

**Table 4.** Indicators of the villager-satisfaction-based evaluation method.

| Primary Indicator | Secondary Indicator | Questions in the Questionnaire (5-Point Likert Scale) | Corresponding Social Factors |
|---|---|---|---|
| Overall satisfaction (1) | Overall satisfaction with tourism development | Are you generally satisfied with the tourism development of the village in recent years? If not, what are the main reasons for dissatisfaction? | Public participation, public opinion, and social satisfaction |
| Satisfaction with physical environment construction (6) | Road and transportation | How are the road and transportation influenced by the tourism development in your village? | / |
| | Eco-environment | How is the ecological environment influenced by the tourism development in your village? | / |
| | Housing conditions | How are the living conditions influenced by the tourism development in your village? | / |
| | Water and electricity supply | Are you satisfied with the village water and electricity supply after the tourism development? | / |
| | Lighting facilities | Are you satisfied with the village lighting facilities after the tourism development? | / |
| | Network communication | Are you satisfied with the village network communication after the tourism development? | / |
| Rural scene (4) | Natural landscape | Does the natural scenery of the village look better after tourism development? | Public participation, public opinion, and social satisfaction |
| | Design of new/reconstructed buildings | Do the buildings in the village look better after tourism development? | Public participation, public opinion, and social satisfaction |
| | Imitation tendency of new/reconstructed design | Is it possible for you to imitate or learn from the new and reconstructed houses when repairing your own house in the future? | Social demonstration |
| | Protection and inheritance of traditional culture | Are you satisfied with the protection and inheritance of traditional culture in the process of tourism development? | Public participation, public opinion, and social satisfaction; social demonstration |
| Economic and social development (3) | Public entertainment activities | How are the public entertainment activities influenced by the tourism development in your village? | Community relations and social progress |
| | Medical and health service | How is the medical and health service influenced by the tourism development in your village? | Community relations and social progress |

Table 4. *Cont.*

| Primary Indicator | Secondary Indicator | Questions in the Questionnaire (5-Point Likert Scale) | Corresponding Social Factors |
|---|---|---|---|
| | Income growth | How has your income changed after tourism development? Are you satisfied with the changes? | Community relations and social progress |
| Social relationship (5) | Relationship with indigenous villagers | How are the relationships among native villagers after the tourism development? | Community relations and social progress; social equity |
| | Relationship with new villagers | What is your attitude towards the new villagers during the process of tourism development? | Community relations and social progress; social equity |
| | Relationship with tourist | What is your attitude towards the tourists during the process of tourism development? | Community relations and social progress; social equity |
| | Performance of village committee | What is your attitude towards the village committee during the process of tourism development? | Community relations and social progress; social equity |
| | Performance of government | What is your attitude towards the government during the process of tourism development? | Community relations and social progress; social equity |

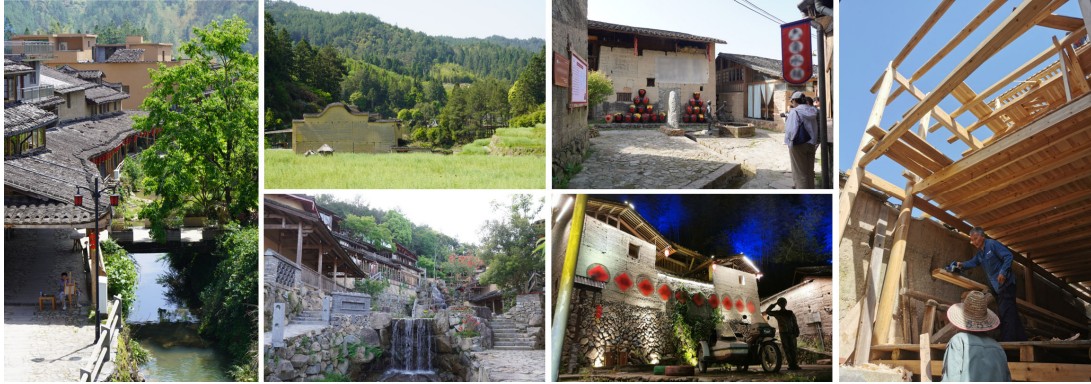

**Figure 7.** Tourist traditional villages in the Jiufeng Mountain area (image source: photographed by the author).

The following factors received special consideration in the formulation process:

- Increase indicators related to social relationship satisfaction. The secondary indicators are refined according to the five groups of indigenous villagers, new villagers, tourists, government personnel, and village committee and village officials. Such indicators can not only directly reflect the harmonious degree of interpersonal relationships, but also reflect the fairness of social distribution.
- Increase indicators related to the rural esthetic satisfaction. For the prevalent problem of traditional dwellings' reconstruction and utilization in the tourism village development, these indicators aim to reflect the social satisfaction of tourism development through the villager satisfaction with the natural landscape, the design of new/rebuilt buildings, and the protection and inheritance of traditional culture. Additionally, the

tendency of imitation in the design of newly constructed or renovated structures was incorporated to investigate the social demonstration of tourism development.

- Simplify the relevant indicators for satisfaction with the physical environment construction. Objective evaluations of the physical environment construction generally align with the subjective attitudes of the villagers: as long as the relevant construction meets objective functional standards, villagers are generally satisfied. Therefore, there is no need to collect additional subjective opinions. Thus, only essential factors for mountainous area tourism development, such as road transportation, ecological environment, living conditions, water and electricity, lighting, and communication, were retained.
- Simplify the phraseology and the size of the questionnaire. Due to the relatively low level of education of elderly villagers, the questionnaire should be easy to understand, and the total number of questions should be controlled at 20–30 and completed in 3–5 min to ensure the initiative of villagers and the quality of answers. The questions applied to the questionnaire are shown in Table 4.

## 4. Data Collection and Result Analysis

### 4.1. Source of Samples and Reliability and Validity Analyses

In July 2023, the research team distributed satisfaction questionnaires to villagers in Beiqian Village and Longtan Village. A total of 58 questionnaires were collected and 53 were valid, including 29 in Beiqian Village and 24 in Longtan Village. All the satisfaction indicators are measured using a 5-point Likert scale in the questionnaire.

#### 4.1.1. Reliability Analysis

Cronbach's reliability coefficient (Cronbach) was used in this paper (Table 5). The data show that the reliability coefficient of the questionnaire was 0.783, greater than 0.7 and close to 0.8. Therefore, it is indicated that the consistency of the answers is relatively high, and the reliability of the survey is acceptable.

**Table 5.** Result of the reliability analysis.

| Sample Size | Item | Cronbach |
|:---:|:---:|:---:|
| 58 | 22 | 0.783 |

#### 4.1.2. Validity Analysis

The Kaiser–Meyer–Olkin test and Bartlett's test of sphericity were used for the validity analysis in this research (Table 6). Ten samples were randomly selected for data calculation, and the KMO value of this study was 0.606, greater than 0.5, which means the high correlation between the score of each question and the total score. At the same time, the *p*-value of Bartlett's test of sphericity was close to 0.000, lower than 0.005, which is a desirable structural validity to the survey.

**Table 6.** Result of the validity analysis.

| Parameter | | Value |
|:---:|:---:|:---:|
| KMO | | 0.606 |
| Bartlett's test of sphericity | $x^2$ | 1018.061 |
| | df | 231 |
| | *p* | 0.000 |

### 4.2. Results of the Villager Satisfaction Evaluation

This research analyzed data through comparative and correlation studies. On the one hand, the satisfaction of various indicators in the two villages were calculated, and on the other hand, the two villages were compared regarding similarities and differences for the same indicator (Figure 8 and Table 7).

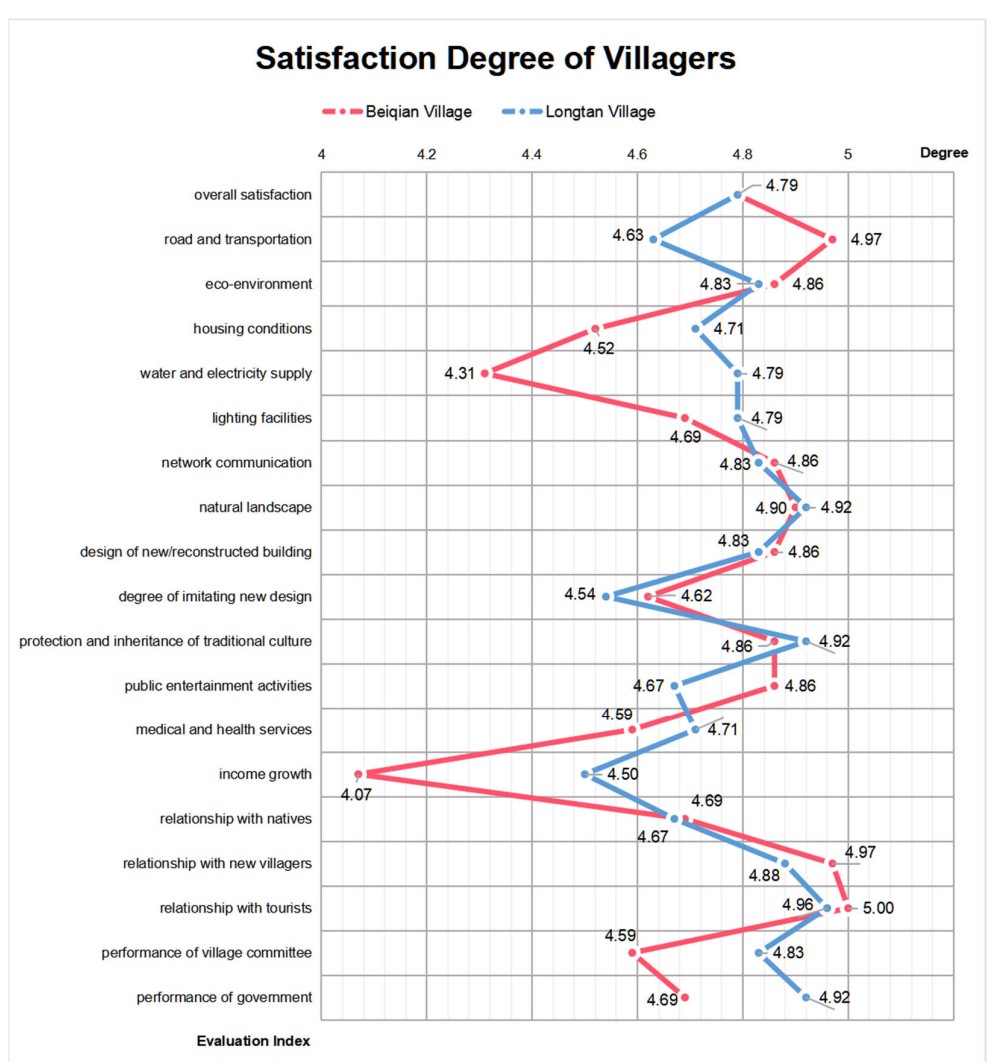

**Figure 8.** Satisfaction degree of the villagers.

First of all, the overall satisfaction of tourism development in both villages was high, with the same average of 4.79, and none of the respondents selected the "dissatisfied" or "very dissatisfied" options. These findings indicate a broad consensus among the villagers regarding their approval of the tourism development initiatives in both villages, signifying relatively favorable social outcomes.

In terms of satisfaction with physical environment construction, the villager satisfaction with the roads and transportation of Beiqian (4.97) was significantly higher than that of Longtan (4.63), while the satisfaction with the water and electricity supply (4.31) and living conditions (4.52) of Beiqian was significantly lower than that of Longtan (4.79 and 4.71). The satisfaction with the ecological environment and network communication of both villages was similarly high.

Regarding the satisfaction with rural esthetics, the natural landscape, the design of new and reconstructed buildings, and the protection and inheritance of traditional culture have been highly appraised by the villagers of the two villages, indicating that the architecture and landscape design in the tourism development process of both villages were appreciated. Meanwhile, the villagers of the two villages exhibited a pronounced inclination to imitate the new and reconstructed buildings (Beiqian, 4.54; Longtan, 4.62).

In terms of economic and social development, Longtan villagers' satisfaction with income growth (4.50) was significantly higher than that of Beiqian Village (4.07), but the

satisfaction with public entertainment activities (4.67) was lower than that of Beiqian Village (4.86). The two villages' satisfaction with medical and health services were similar.

**Table 7.** Satisfaction degree analysis.

| Evaluation Index | Longtan Village (s1) | Beiqian Village (s2) | Degree of Difference (d) [1] |
|---|---|---|---|
| Overall satisfaction | 4.79 | 4.79 | 0.000 |
| Road and transportation | 4.63 | 4.97 | 0.073 |
| Eco-environment | 4.83 | 4.86 | 0.006 |
| Housing conditions | 4.71 | 4.52 | 0.042 |
| Water and electricity supply | 4.79 | 4.31 | 0.111 |
| Lighting facilities | 4.79 | 4.69 | 0.021 |
| Network communication | 4.83 | 4.86 | 0.006 |
| Natural landscape | 4.92 | 4.9 | 0.004 |
| Design of new/reconstructed buildings | 4.83 | 4.86 | 0.006 |
| Degree of imitation for new designs | 4.54 | 4.62 | 0.018 |
| Protection and inheritance of traditional culture | 4.92 | 4.86 | 0.012 |
| Public entertainment activities | 4.67 | 4.86 | 0.041 |
| Medical and health services | 4.71 | 4.59 | 0.026 |
| Income growth | 4.5 | 4.07 | 0.106 |
| Relationship with the natives | 4.67 | 4.69 | 0.004 |
| Relationship with the new villagers | 4.88 | 4.97 | 0.018 |
| Relationship with tourists | 4.96 | 5 | 0.008 |
| Performance of the village committee | 4.83 | 4.59 | 0.052 |
| Performance of the government | 4.92 | 4.69 | 0.049 |

[1] Degree of difference (d) shows the relative difference of the same indicator between the two villages. d = |s1 − s2|/min {s1, s2}. If the d value is less than 0.03, it is considered similar, and if the d value is greater than 0.03, it is considered significantly different.

Considering social relationships, Longtan villagers were more satisfied with the government (4.92) and village committee (4.83), while Beiqian villagers were less satisfied (4.69 and 4.59). The villagers of both villages had a similar satisfaction level with the relationship with indigenous villagers and are welcoming towards tourists and new villagers.

## 5. Discussion

### 5.1. Validity Analysis of the Villager Satisfaction Evaluation Method

Based on the above experimental results and the actual situation of the two villages and their tourism development trajectories, it was demonstrated that the tourism village development evaluation method proposed in this paper has a certain degree of validity. This validity is primarily reflected in two aspects. On the one hand, the quantitative results of the questionnaire evaluation are consistent with what the research team learned from the interviews with indigenous villagers during the fieldwork. For example, the residents of both villages exhibit a high level of hospitality and a welcoming attitude towards tourists and new villagers. During the visit, as "tourists", the research team distinctly felt the warmth and enthusiasm from the local villagers. When discussing the "new villagers", indigenous villagers also expressed appreciation for the homestays and other investment projects initiated by the "new villagers". Encouraged by them, the indigenous villagers are eager to try similar ventures themselves. Especially, some of the villagers who work in the city think it is a better choice to return to the village for entrepreneurship if the income is comparable to or even slightly lower than working outside. In terms of villagers' income satisfaction in the questionnaire, satisfaction in Beiqian Village was significantly lower than that in Longtan Village. During the actual visit, Beiqian Village was obviously more deserted, and many tourism projects were in a half-closed state, because most of the projects in Beiqian Village are held by the government and relevant tourism companies with higher operating costs, while the indigenous villagers are unable to obtain direct income from tourism development, and the limited flow of visitors is unable to drive enough consumption to support the villagers'

entrepreneurial projects. Longtan Village, on the other hand, was significantly more bustling, with a large number of indigenous villagers using the idle space of their homes to operate small-scale entrepreneurial projects, such as tearooms, bars, homestay accommodations, and kiosks, which have flexible opening hours and almost no fixed operating costs; therefore, the projects can continue to operate, resulting in a significantly better visitor experience than that of Beiqian Village, as well as attracting a greater customer flow, which in turn boosts the tourism revenues of the entire region.

On the other hand, the evaluation method reflects the differences in tourism development paths of different villages through refined social level indicators. In terms of villager satisfaction with social relationship, the indigenous villagers in Longtan Village were more satisfied with government departments and village committees than those in Beiqian Village. The reason for this may lie in the fact that, although the government has played a leading role in the tourism development of both villages, the government in Longtan Village has been more restrained in its involvement, and the logic behind policy implementation has been more transparent. The village committee has also established a harmonious and trusting relationship with both indigenous and new villagers through systemic innovations, such as the "Gong Liao method". In contrast, despite the government's greater financial support for Beiqian Village, indigenous villagers are less involved in rural settlement construction and the planning of tourism activities, and even the interviewed village carders did not understand the reasons and the importance of the traditional buildings' protection policies. However, in terms of the degree of imitation for new or renovated buildings, Beiqian Village surpasses Longtan Village. This may be due to the fact that Longtan Village adopted the "old house renting" model, in which the renovation of houses is led by the personal preferences of the tenants, resulting in a wide variety of architectural styles. On the other hand, the houses of Beiqian Village are based on the government-led top-down model, emphasizing a stronger overall and coordinated appearance, along with a better functionality and quality. Therefore, it lends itself better to imitation and is well-received.

In summary, in the selected villages, the objective quantitative results obtained in this study coincide with the subjective opinions collected during the field visits, reflecting the effectiveness of the evaluation method in collecting the real opinions of indigenous villagers, and the "indigenous villagers' opinions" also correspond to the specific conditions of the tourism village development, which highlights the essential value of the "indigenous villagers' opinions" in tourism development.

*5.2. The Process of Introducing Villager Satisfaction into the Evaluation Method*

In addition to proposing the villager-satisfaction-based evaluation methods for tourism village development in mountainous areas, the generation process is also worthy of reference for the same type of research. The process of proposing and applying the evaluation method in this paper was as follows: ① Formulate the question: How can the opinions of villagers be effectively and adequately expressed? ② Focusing on the research subject: Mountainous tourist villages face challenges during development and post-operation stages. ③ Theoretical research: Summarize the classification logic and the pros and cons of the commonly used evaluation indicators in existing studies, serving as the theoretical foundation. ④ Experiential research: Gather insights from interviews to understand villagers' perspectives on local rural tourism development, providing an empirical basis. ⑤ Design the evaluation indicator system: Building upon the theoretical and empirical foundations, propose primary and secondary evaluation indicators and transform them into implementable questionnaires. ⑥ Correlation analysis: Compare questionnaire results with the actual village conditions to verify the feasibility of the evaluation method. ⑦ Address the issue: Provide research findings as feedback to the leaders of village collectives and the government departments responsible for rural tourism development, thereby enhancing the villagers' voice in the process.

The above process was organized around the core concept of "villager satisfaction" and achieved a complete research loop of the exploration of its meaning, theoretical and empirical research, experimental application, and feedback from case studies. Regarding this practical issue, we achieved a dual exploration of both theoretical and applied methods.

*5.3. Application Scenarios of the Villager Satisfaction Evaluation Method*

In fact, in addition to being a means of "post-occupancy evaluation", there are various potential application scenarios for the "villager satisfaction" method. For example, conducting villager satisfaction surveys at different stages of tourism development can provide a better understanding of villagers' attitudes, opinions, and feedback regarding tourism development. During the demonstration and planning stages of tourism development, a survey about the existing tourism development examples can be conducted among villagers. This helps to identify their main concerns of tourism development and allows the optimization of tourism project planning based on the feedback collected. This ensures that tourism projects receive community participation and support from the very beginning. Tourism village development is limited by the scale of investment and is generally progressive. Therefore, it is advisable to periodically organize community meetings or collective discussions during the tourism development process, conducting surveys on villager satisfaction. This helps to adjust and improve the project in a timely manner and to avoid the expansion of hidden dangers or the intensification of conflicts. After tourism development has become relatively mature, it is more necessary to conduct villager satisfaction evaluations to summarize the experience and assess the effectiveness to provide a reference for future development.

Of course, it is important to note that different development stages, paths, and types of rural areas will require tailored evaluation criteria based on field investigations to more accurately and authentically reflect the villagers' opinions.

**6. Conclusions**

The main concern of this paper was the presentation of the theoretical research and questionnaire design of the evaluation methodology. Based on two specific mountainous villages, and based on the literature research and field surveys in the Jiufeng Mountain area, an evaluation method based on villager satisfaction, containing 5 primary indicators and 19 secondary indicators, was designed and transformed into a questionnaire provided to the villagers.

The application of this questionnaire in Beiqian Village and Longtan Village in the Jiufeng Mountain area confirmed the distinct characteristics of tourism development and development models in the two villages. This, to some extent, validates the scientific and effective nature of the evaluation method. This paper presented a comprehensive research cycle that begins with practical issues and incorporates theoretical and empirical research. It also balances macro-scientific methods with specific analysis and application. Furthermore, the chosen research subjects had a certain level of representativeness within the context of China, making this research particularly relevant and practical for addressing current issues in rural development and tourism village development in China.

**Author Contributions:** Conceptualization, N.L.; Methodology, N.L.; Software, X.Z.; Investigation, N.L. and K.Y.; Resources, N.L.; Data curation, K.Y. and X.Z.; Writing—original draft, N.L., K.Y. and X.Z.; Writing—review & editing, N.L.; Visualization, N.L.; Project administration, N.L. All authors have read and agreed to the published version of the manuscript.

**Funding:** This research was funded by [National Natural Science Foundation of China] grant number [52278022] and The APC was funded by [National Natural Science Foundation of China].

**Institutional Review Board Statement:** Not applicable.

**Informed Consent Statement:** Informed consent was obtained from all subjects involved in the study.

**Data Availability Statement:** Data sharing not applicable. No new data were created or analyzed in this study. Data sharing is not applicable to this article.

**Conflicts of Interest:** The authors declare no conflict of interest.

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
