# Peer review of "Ignored Opinions: Villager-Satisfaction-Based Evaluation Method of Tourism Village Development—A Case Study of Two Villages in China"

_sustainability, doi:10.3390/su152215726_

Round 1

Reviewer 1 Report

Comments and Suggestions for Authors

line 161: Table 1;

line 215: Table 2;

line 238: Table 3;

line 248: Table 4;

line 304: indicators are written in too light gray;

In the Table 1. (actualy 2) “Common Evaluation indicators in Research on the Evaluation of Rural Human Settlements”, it does not seem correct to place the same evaluation indicators as Primary and Secondary indicators. Secondary indicators should be identified in the case of “social cultures” and “economic development”.

Figure 6.:  how are indicator values established? Is there a threshold value below which interviewee' opinion is considered negative?

Reviewer 2 Report

Comments and Suggestions for Authors

The paper is very good, but the main issue is that it addresses almost exclusively to a Chinese public, it has almost no relevance for other researchers. This is somehow reflected in the conclusions, where the authors themselves describe the paper as relevant for China (only), and in the list of references, which includes only Chinese authors. It is interesting to note that some non-Chinese authors are mentioned throughout the text (Pena, Caudill, Preiser, and more) but they are not quoted as they should, and they are not mentioned in the list of references.

Comments on the Quality of English Language

There are several mistakes in terms of the English language:

p. 2 line 83: "is the main components", it should be "is the main component"

p. 3 line 139: "latelier" is not a word in English, the whole sentence should be rephrased

p. 4 line 154: Figure 2 "The scope of Jiufeng Mountain Area". I don't really understand what is the meaning of the "scope" here

p. 8 line 225: "froma" are two words, "from a"

p. 8 line 232: "village cadres" - I have a problem with the word "cadres", which appears several times in the paper. I can think that it means "the village leaders" or perhaps the representatives of the Communist Party in the villages. However, I would change the word for something easier to understand by the readers.

p. 11, the top two questions are wrongly formulated in English: "Do the village's natural scenery looks better?" and "Do the village's buildings looks better"? In the first case, it should be "Does the natural scenery of the village look better"? and in the second case, "Do the buildings in the village look better"?

p. 13, line 306 "the overall satisfactions... are high" should be "the overall satisfaction... is high"

p. 13 line 311 "Villager satisfaction" should be "villager satisfaction", as it is not at the beginning of a sentence

p. 13 lines 314-315 "Satisfactions... are high" should be "Satisfaction... is high"

And there are more instances where the word "satisfactions" appear - but this is wrong, in English the word "satisfaction" does not usually have a plural form.

p. 15 line 411 "villager satisfaction - missing the end of the quotation mark

p. 15 lines 414-415 "Survey" should be "a survey", "to villagers" should be "among villagers", and then comes a larger space

p. 15 line 419 "it's advisable" is not necessarily wrong, but it should not be used in a scientific paper. "it is advisable" is the correct version

p. 16 line 426 "it's important" - the same thing, "it is important"

p. 16 line 435 "North Qian Village" - I thought the questionnaire was applied in Beiqian Village??

Reviewer 3 Report

Comments and Suggestions for Authors

Interesting article. The main objective of the research, to know what the locals think about tourist activity, is not restricted to China, is a situation basically in the rest of the world. I like the methodology to start embracing the problem. However, I found it a little bit limited to China. What I mean, is that there are other similar research conducted in other areas and the authors do not cite them or use them as case example. What makes China different or what are similar social/cultural contexts? is it because is totally China state-oriented?

It is also important not only to know the local's opinion about the tourist activity, but also how this activity changes the local landscape, and not always for good.

In this regard, I found interesting in the survey that when the locals were asked if they were ok with the building renovations, they mentioned that as long they met functional standards it was fine. The authors mention that they do not see the need to ask further questions!! Here we see a clear case of tourist activity disrupting the traditional local/rural architecture and is important to follow the truth. If one of the reasons this village is getting tourist interest is the cultural patrimony (aka local traditional architecture) what is going to happen if that patrimony disappears? 

In line 192-194, the authors mentions that in 2022 Longtan village got 300, 000 visitors!!! I did not see any mention of if that is good or bad for the village, only the sense that "is good for business", sell at any cost. Is that the objective of the tourist activity in the region? Is there, not a capacity pressure analysis? What about the local resources? 

I think the article had good intentions but needs to be a little more impartial and critical about the research and results.

Reviewer 4 Report

Comments and Suggestions for Authors The topic is current and very relevant.

The research question is not very clear.

The methodology does not exist.

Some figures are never mentioned in the text.

The literature review must be a chapter and not a subchapter of Introduction.

A reorganization of the paper would benefit the clarity of the same.

Round 2

Reviewer 4 Report

Comments and Suggestions for Authors

I am satisfied with the changes made. The paper was clearer and better.